# Genetic effects on variability in visual aesthetic evaluations are partially shared across visual domains

Giacomo Bignardi [1,2✉], Dirk J. A. Smit [3], Edward A. Vessel [4,5], MacKenzie D. Trupp[6,11], Luca F. Ticini[7], Simon E. Fisher [1,8] & Tinca J. C. Polderman [9,10]

The aesthetic values that individuals place on visual images are formed and shaped over a lifetime. However, whether the formation of visual aesthetic value is solely influenced by environmental exposure is still a matter of debate. Here, we considered differences in aesthetic value emerging across three visual domains: abstract images, scenes, and faces. We examined variability in two major dimensions of ordinary aesthetic experiences: taste-typicality and evaluation-bias. We build on two samples from the Australian Twin Registry where 1547 and 1231 monozygotic and dizygotic twins originally rated visual images belonging to the three domains. Genetic influences explained 26% to 41% of the variance in taste-typicality and evaluation-bias. Multivariate analyses showed that genetic effects were partially shared across visual domains. Results indicate that the heritability of major dimensions of aesthetic evaluations is comparable to that of other complex social traits, albeit lower than for other complex cognitive traits. The exception was taste-typicality for abstract images, for which we found only shared and unique environmental influences. Our study reveals that diverse sources of genetic and environmental variation influence the formation of aesthetic value across distinct visual domains and provides improved metrics to assess inter-individual differences in aesthetic value.

[1] Language and Genetics Department, Max Planck Institute for Psycholinguistics, Nijmegen, the Netherlands. [2] Max Planck School of Cognition, Stephanstrasse 1a, Leipzig, Germany. [3] Department of Psychiatry, Amsterdam UMC, Amsterdam, The Netherlands. [4] Max Planck Institute for Empirical Aesthetics, Frankfurt, Germany. [5] Department of Psychology, City College, City University of New York, New York, NY, USA. [6] Department of Cognition, Emotion, and Methods in Psychology, University of Vienna, Vienna, Austria. [7] Department of Psychology, Webster Vienna Private University, Vienna, Austria. [8] Donders Institute for Brain, Cognition and Behaviour, Radboud University, Nijmegen, The Netherlands. [9] Clinical Developmental Psychology, Faculty of Behavioural and Movement Sciences, Vrije Universiteit Amsterdam, Amsterdam, The Netherlands. [10] VKC Psyche, Child and Adolescent Psychiatry and Psychosocial Care, Amsterdam UMC, Amsterdam, The Netherlands. [11] Present address: Department of Cognitive Neuroscience, Donders Institute for Brain, Cognition and Behaviour, Radboud University Medical Center, Nijmegen, The Netherlands. ✉email: giacomo.bignardi@mpi.nl

Both the long history of aesthetics[1,2] and the recently renewed interest in this field[3–9] have led to substantial discoveries in how humans evaluate sensory experiences. These discoveries revealed organisational principles which guide the formation of visual aesthetic value, from the value assigned to ordinary sensory experiences to visual art[10–13] to corresponding neural correlates[14–20] (see[21] for a review) and have informed recent computational models of how aesthetic value comes to be[22–24]. One such emerging principle indicates that inter-individual differences in aesthetic evaluations are the norm rather than the exception[25]. People tend to display different aesthetic sensitivity towards features of the stimuli being evaluated[26,27], and differ in the extent to which they are open to, and derive pleasure from, aesthetically rewarding experiences[28–30]. Further, people show a great degree of variation in individual taste[11,13,31] even for stimuli for which aesthetic evaluations are mostly agreed upon, such as faces[11,32,33].

Although extensive research has investigated such inter-individual differences, with recent discoveries starting to shed light on major underlying dimensions of such variability[13], one question remains overlooked: what are the sources of differences in aesthetic evaluations? Many studies suggest that aesthetic value is formed and shaped by prior experiences in individuals, groups and societies[12,31,34–38]. However, complementary evidence challenges the idea that the environment alone is the only source of variation. For example, evidence from behavioural genetic studies shows that genetic predispositions make substantial contributions to variation in attitudes, interests and engagement toward music and arts[39–42], proneness to instances of aesthetic experiences, such as aesthetic chills[43], and even major dimensions of cultural taste and participation[44].

Yet, little is known about the extent to which genetic variation contributes to what makes subjective aesthetic evaluations differ between individuals. Only a few studies have systematically tested whether the formation of aesthetic value is caused by environmental exposure alone or if genetic predispositions also constrain variation in aesthetic evaluations[32,45–48]. These studies employed the Classical Twin Design (CTD[49,50]), a widely used approach to quantify genetic and environmental contributions to variation. The CTD compares monozygotic (MZ) twin pairs, who share the same fertilised egg and are roughly genetically identical, to dizygotic (DZ) twin pairs, who are derived from two fertilised eggs and are on average 50% genetically identical (with respect to allelic variation). Based on the extent to which MZ co-twins are more similar to each other than DZ co-twins are on a given trait, it is possible to quantify the amount of variability that correlates with additive genetic factors, also known as heritability ($h^2$). We note here that although $h^2$ is a parameter capturing the association between genetic variants and phenotypic variability in a population, it does not allow for causal or deterministic inference[51]. Moreover, we stress that $h^2$ is both population- and environment-specific, as allelic frequencies within a population and different environmental conditions can change the amount of genetic and environmental variance, influencing the final $h^2$ estimate (we refer the interested reader to excellent prior published work discussing concepts and misconceptions of heritability in-depth[52]).

Using the CTD, two studies[47,48] examined the extent to which idiosyncrasies in aesthetic judgements were in line with experts' opinions for images of paintings and drawings. The authors found contradictory results, with $h^2$ estimates for aesthetic judgements ranging from 0%[48] to 67%[47], a broad spread that is likely related to small sample sizes, making it difficult to draw reliable conclusions (59 and 57 Italian and North American same-sex twin pairs in the first and 61 Italian same-sex twin pairs in the second, respectively). Three other studies from Zietsch et al.[46],

Germine et al.[32] and Sutherland et al.[45] investigated larger twin samples focusing on facial aesthetic preferences. Heritability estimates were 33% for specific preferences for dimorphic male traits (i.e., the masculinity of the face being evaluated)[46], and 22% and 30% for more general individual preferences for faces (2160 Finnish female twins and siblings, 796 Australian same-sex twin pairs, and 1264 Australian same-sex twins, respectively)[32,45]. Nevertheless, a description of the extent to which genetic or environmental influences impact the formation of variation in aesthetic evaluations beyond individual preferences of faces is still lacking.

Here, we extend previous work by investigating the etiology of foundational metrics of aesthetic evaluations of three image domains, namely abstract images, scenes, and faces, with a complementary re-analysis for the latter domain. We also go beyond a pure description of univariate etiological sources of variation on aesthetic evaluations by examining possible shared genetic and environmental etiological sources that exert their influences across different visual domains.

First, our analyses revealed that aesthetic evaluations for abstract images, scenes and faces consistently elicited a diversified range of reliable individual preferences. Individuals expressed the most dissimilar (idiosyncratic) preferences for abstract images, intermediate similarity for faces, and most similar (shared) preferences for scenes. We then quantified the extent to which genetic and environmental factors influence inter-individual variation in aesthetic evaluations. We compared the extent to which pairs of individuals, including pairs of twins, differed in their preferences. This allowed us to answer whether familial genetic relatedness relates to variability in aesthetic preferences. We then analysed the sources of variation in the primary dimensions of inter-individual differences in aesthetic value[46]. Specifically, we examined how typical individuals' aesthetic evaluations were compared to the group average (taste-typicality[13]), which has been recently claimed to be the primary dimension along which inter-individual differences in aesthetic values vary[13]. We complemented our analyses by describing the sources of variation in the overall aesthetic value expressed by individuals, which we refer to as evaluation-bias, a largely neglected metric in many contemporary studies assessing aesthetic evaluations[13,32,33]. Finally, based on evidence suggesting that aesthetic value is represented in both a domain-general and a domain-specific manner[18], we explored the degree to which etiological influences are shared across the three image domains. While doing so, we distinguished genetic and environmental influences on the observed covariation between taste-typicality or evaluation-bias across visual domains from the amount of genetic and environmental correlations (see ref.[53] for details). All measures and analyses were tested for robustness and confounders using a genetically informative validation sample with additional data on aesthetic evaluations[45].

## Results
**Inter-individual differences in visual aesthetic evaluations.** We re-analysed data originally obtained by Germine et al.[32], publicly available at https://osf.io/c3hz6/[54], comprising a discovery sample of 1115 MZ and 432 DZ twins, mean age 45 y (sd = 13 y, ranging from 21 to 68 y). To test the validity and replicability of our results, we also re-analysed data from a validation sample collected by Sutherland et al.[55], publicly available at https://osf.io/35zf8/?view_only=e76c6755dcea4be2adc5b075cae896e8, comprising 815 MZ and 416 DZ twins, mean age 47 y (sd = 15 y, ranging from 16 to 80 y, see Table 1). The twins had originally rated 65 abstract images in the discovery sample, including a subset of 15 repeated images. They had also rated 65 and 74

images of scenes (including 15 and 24 repeats), and 260 and 150 images of faces (60 and 50 repeats) in the discovery and validation samples, respectively (see Fig. 1a and "Methods" for details). Before analysis, we estimated intra-individual reliability ($R_{xx\text{-}intra}$) by calculating the Pearson correlation between repeated ratings for the same images within individuals and excluded participants for whom this reliability was <0.5 (following[11,13]; also intra-image reliability, $R_{xx\text{-}image}$, was >0.5 for all images, see Supplementary Notes and Supplementary Figs. 1 and 2). We also excluded participants who showed no variation in ratings within domains,

**Table 1 Samples of monozygotic (MZ) and dizygotic (DZ) twin pairs per visual domain.**

|  | MZ male | MZ female | DZ male | DZ female |
|---|---|---|---|---|
| All | 133 (133) | 425 (424) | 54 (54) | 162 (162) |
| All (val.) | 126 (94) | 350 (245) | 37 (23) | 219 (137) |
| Abstracts | 130 (103) | 420 (360) | 54 (41) | 161 (145) |
| Abstracts (val.) | – (–) | – (–) | – (–) | – (–) |
| Scenes | 132 (123) | 425 (419) | 54 (51) | 162 (159) |
| Scenes (val.) | 126 (91) | 350 (245) | 37 (22) | 219 (137) |
| Faces | 133 (122) | 425 (407) | 54 (51) | 162 (158) |
| Faces (val.) | 125 (89) | 340 (232) | 37 (21) | 216 (128) |

Top two rows show original samples[32,45] before exclusion. Remaining rows show the final numbers of twin pairs after applying exclusion criteria for each visual domain. The number of complete pairs is shown between parentheses. Validation sample (i.e., Sutherland et al.[45] sample).
Val.: validation.

following Germine et al.[32] (Table 1 gives the final sample). Variance Partitioning Coefficients (VPC[45,56]) extracted from multi-level models fitted to ratings of only one twin per pair (to avoid confounding familial resemblances, see "Methods" and Supplementary Note and Supplementary Fig. 3) confirmed that images elicited a remarkably diversified range of stable inter-individual differences in aesthetic evaluations, with the sum of individual-level VPCs capturing 69%, 35% and 44% of the total variance within abstract images, scenes and faces, respectively (Fig. 1b). VPCs capturing the effect of image repetition (i.e., exposure) indicated small systematic effects (~1%) on the overall variance in aesthetic ratings (but see Supplementary Note 3 for a detailed discussion). Thus, for subsequent analyses, we used the average ratings of the repeated images within participants to increase the signal-to-noise ratio of the ratings.

**Measures of inter-individual differences in aesthetic evaluation.** To quantify individual aspects of aesthetic value, we computed pair- and individual-level metrics (Fig. 1c). The pairwise aesthetic agreement was calculated as the inter-individual correlation ($r_{inter}$) of the ratings per domain within each MZ, DZ, and matched unrelated pair (UR) classes (adapted from[31]). UR comprised two same-sex, familial unrelated individuals selected from the twin sample and matched to minimise their age differences (see "Methods"). We quantified how typical individuals' aesthetic evaluations were compared to the group average, which we refer to as taste-typicality, by computing the mean minus two ($mm2$; adapted from ref. [11]). The $mm2$ was calculated by

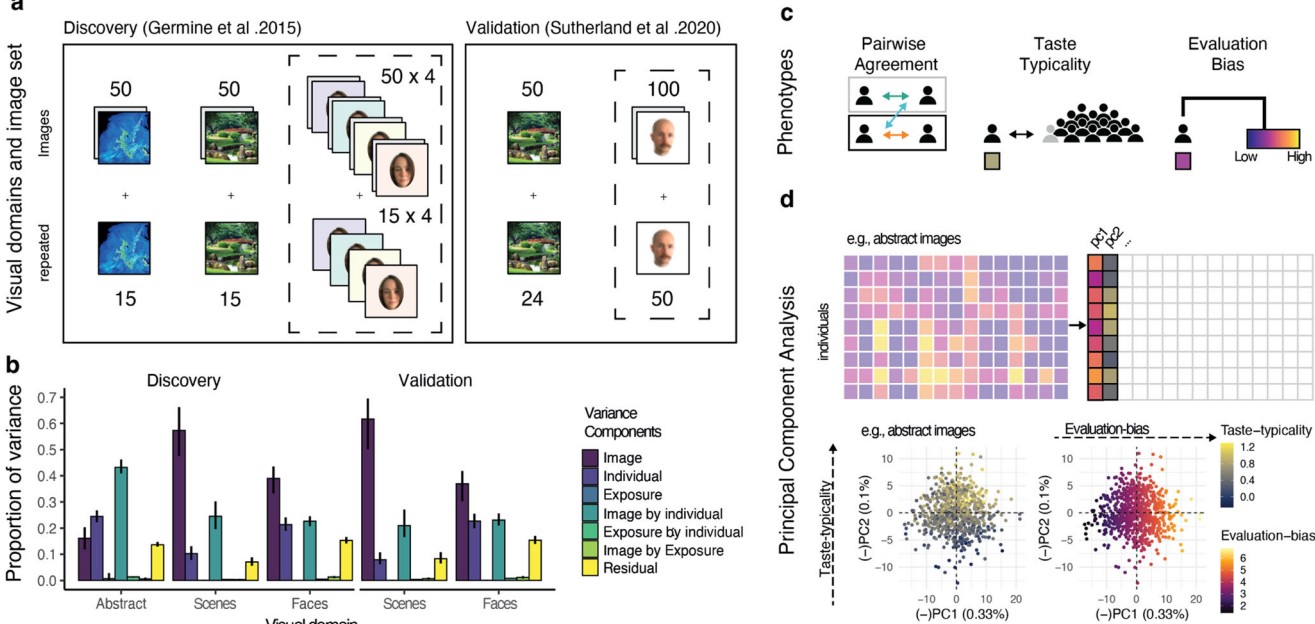

**Fig. 1 Inter-individual differences in aesthetic evaluations. a** Examples of stimuli representing images used by Germine et al.[32] and Sutherland et al.[45]. The authors originally investigated the etiology of individual preferences for faces (dashed box) but also collected data on abstract images and scenes. Examples of abstract images and images of a scene were obtained from ref. [31]. Images of faces were created as an example of the images of faces. Individuals depicted in the images gave informed consent for publication. The exact images used by both authors are not shown (but see "Methods" for details). **b** Multilevel modelling for twins' (one per pair only) ratings of abstract images and images of scenes and faces (results for all twins are shown in Supplementary Fig. 3). The variance coefficients quantify the amount of variance in ratings due to the image (hence shared among individuals), individual features (therefore unique to the individual), the exposure of the images (hence due to the repetition), and their interactions. Error bars represent 95% confidence intervals (CI). **c** The pair-level and individual-level metrics were used to quantify the three aesthetic phenotypes: pairwise agreement, taste-typicality and evaluation-bias. Pairwise agreement reflects the agreement within a MZ (light green), DZ (orange) or unrelated pseudo-randomised (UR; turquoise) pair; Taste-typicality reflects how similar an individual set of ratings is in comparison to the average ratings of everyone else but their twin. Evaluation-bias reflects the overall aesthetic value evoked by a domain of visual images. **d** Results from the PCA over the individual ratings for abstract images. PC1 and PC2 individual scores are plotted in the bottom panel. Colour represents the taste-typicality (left) and evaluation-bias (right) scores.

correlating all the ratings given by one twin per domain with the average ratings of every other individual, excluding the individual's twin (see "Methods" and Supplementary Note and Supplementary Fig. 4 for a detailed discussion on taste-typicality and how it relates to previous metrics used in the literature). The overall aesthetic value evoked by each domain in each participant, which we refer to as evaluation-bias, was computed as the intra-individual average of each participant's ratings (Fig. 1c). To avoid violating assumptions of statistical tests carried out later on taste-typicality and evaluation-bias data, we additionally excluded seven pairs (3 UR and 4 MZ pairs) and six pairs (3 UR, 2 DZ and 1 MZ pair) from the discovery and the validation sample with extreme pairwise ratings. We also excluded two additional individuals with extreme outlying taste-typicality scores (one for faces, one for scenes) from the discovery sample and three individuals with extreme outlying taste-typicality scores for faces in the validation sample.

To confirm that these individual-level metrics captured a substantial proportion of inter-individual variability in aesthetic evaluation, we computed a Principal Component Analysis (PCA) of the individual ratings for each domain. PCA was run only on the first members of each pair to avoid familial confounding. We identified two major axes of variability in the aesthetic evaluation of images, which jointly explained 44%, 41%, and 45% of the variance in ratings for abstract images and images of scenes and faces, respectively. The Pearson correlations between the individual scores extracted from the first PC and the evaluation-bias measure were all $r > 0.99$ in the discovery sample. Correlations between the individual scores extracted from the second component and taste-typicality Fisher z transformed values (see "Methods") were $r(701) = 0.66$, 95% CI [0.62, 0.7] for abstract images, $r(759) = 0.71$, 95% CI [0.68, 0.75] for scenes and $r(752) = 0.72$, 95% CI [0.68, 0.75] for faces (all $p < 0.001$). Results were replicated in the validation sample, with correlations between the first PC and evaluation-bias scores being all $r > 0.99$, and correlations between the second PC and taste-typicality equal to $r(584) = 0.66$, 95% CI [0.62, 0.71] for scenes, and $r(560) = 0.63$, 95% CI [0.57, 0.67] for faces (all $p < 0.001$). Correlations were similar for second twin members (see Supplementary Note and Supplementary Fig. 5 for details; note that for ease of interpretation, when needed, all PCs were flipped to display positive correlations with evaluation-bias and taste-typicality). These findings indicate that taste-typicality and evaluation-bias scores relate to the major dimensions of inter-individual differences in aesthetic value (Fig. 1d).

**Monozygotic twins show higher pairwise aesthetic agreement than dizygotic twins and unrelated pairs.** To investigate genetic contributions, we used the discovery sample and assessed the extent to which pairwise aesthetic agreement differed across MZ, DZ and UR pairs. An ANOVA (type III $3 \times 3$; domain X pair class) carried out on Fisher z-transformed values ($z_{inter}$, see "Methods") revealed a significant, albeit small, effect of pair class on pairwise aesthetic agreement, $F(2, 4242) = 101.88$, $p < 0.001$ ($\eta_p^2 = 0.05$; 95% CI [0.03, 0.06]; results were robust to inclusion of outliers, see Supplementary Note 6). Marginal pairwise agreement averaged across visual domains was highest for MZ, followed by DZ and UR pairs, $r_{inter} = 0.64$, 95% CI [0.63, 0.65]; $r_{inter} = 0.6$, 95% CI [0.58, 0.61]; and $r_{inter} = 0.54$, 95% CI [0.53, 0.55] (Fig. 2a, b, all $p < 0.001$, Bonferroni corrected).

Additional posthoc comparisons revealed that pairwise agreement differences were significant for all pair classes (i.e., pairwise agreement MZ > DZ > UR within each domain, all $p < 0.05$), except for differences between MZ and DZ for

faces, for which the difference was not significant ($p = 0.09$). Effect sizes and CI were all above 0, ranging from $d = 0.21$, 95% CI [0.05, 0.37], for the observed MZ > DZ pair classes difference for faces, to $d = 0.86$, 95% CI [0.74, 0.98], for the MZ > UR pair classes difference for abstract images. Analysis of the validation sample revealed similar results (Fig. 2c, d and Supplementary Figs. 6 and 7), consistent with the directionality of all the effects reported in the discovery sample. One exception was the MZ and DZ pairwise agreement for faces, for which the difference was significant ($p < 0.001$; $d = 0.43$, 95% CI [0.23, 0.63]). Thus, under the assumptions of the CTD, results indicated that genetic factors play a significant, albeit small, role in preferences for visual images. Moreover, results indicated that unrelated familial individuals are less similar in their aesthetic preferences than related ones.

**Univariate CTD estimates genetic effects for all dimensions of aesthetic value except taste-typicality for abstract images.** To quantify and describe genetic and environmental sources of variation in aesthetic evaluations, we compared similarities across MZ and DZ twins' taste-typicality and evaluation-bias for abstract images and images of scenes and faces. The phenotypic variance in taste-typicality and evaluation-bias was partitioned into genetic (additive [A]) and environmental (shared or common [C] or unique environmental [E]) variance components by performing univariate Structural Equation Modelling (SEM) of the twin data[57], using the direct symmetric approach[58] (see "Methods"). As unsystematic measurement error can inflate the E component and thus deflate both A and C estimates, we also confirmed that reliabilities of taste-typicality and evaluation-bias were good to excellent (both within and between days of testing, see Supplementary Notes 7 and 8). Exclusion criteria matched those reported above for PCA. Means, phenotypic and twin correlations, and 95% CI were extracted from the saturated model, which included sex and age as covariates. Homogeneity of means and variances across twin order and zygosity were met, as tested by comparing the saturated model and the most parsimonious models with equated means and variances (reported in Supplementary Data 1). Based on the pattern of twin correlations and 95% CI between DZ and MZ twins[59] (Fig. 3a–d, see Supplementary Table 1 for details), either AE or ACE models were expected to account for variance in all traits for faces and scenes. For evaluation-bias for abstract images, either AE or ADE models were expected to best account for variance, while for taste-typicality for abstract images, a CE model was expected. None of the parsimonious ACE or ADE models were found to significantly decrease model fit when compared to the full saturated model (all differences in $\chi^2(\Delta df = 6)$, $p > 0.05$).

Further model comparisons confirmed that AE models were the best fitting models for taste-typicality (scenes and faces ACE vs AE, $\chi^2(\Delta df = 1)$, $p > 0.05$) and evaluation-bias data (all ACE vs AE, $\chi^2(\Delta df = 1)$, $p > 0.05$), for all stimulus domains, except for taste-typicality of abstract images (ACE vs AE, $\chi^2(\Delta df = 1) = 5.304$, $p = 0.02$). Genetic influences (A) explained 36% ($h^2$ 95% CI [0.29; 0.43]) of the variance in taste-typicality for images of scenes, and 32% ($h^2$ 95% CI [0.24; 0.4]) and 26% ($h^2$ 95% CI [0.18; 0.33]) of the variance in evaluation-bias for abstract images and scenes. Estimates were similar to those obtained for faces, except for variation in taste-typicality for abstract images, which was entirely explained by environmental influences, with 29% (95% CI [0.22; 0.36]) of variance being explained by shared environmental factors (C). Consistent results were observed in the validation sample (see Table 2 for full and restricted models estimates; a comprehensive summary is reported in Supplementary Data 2).

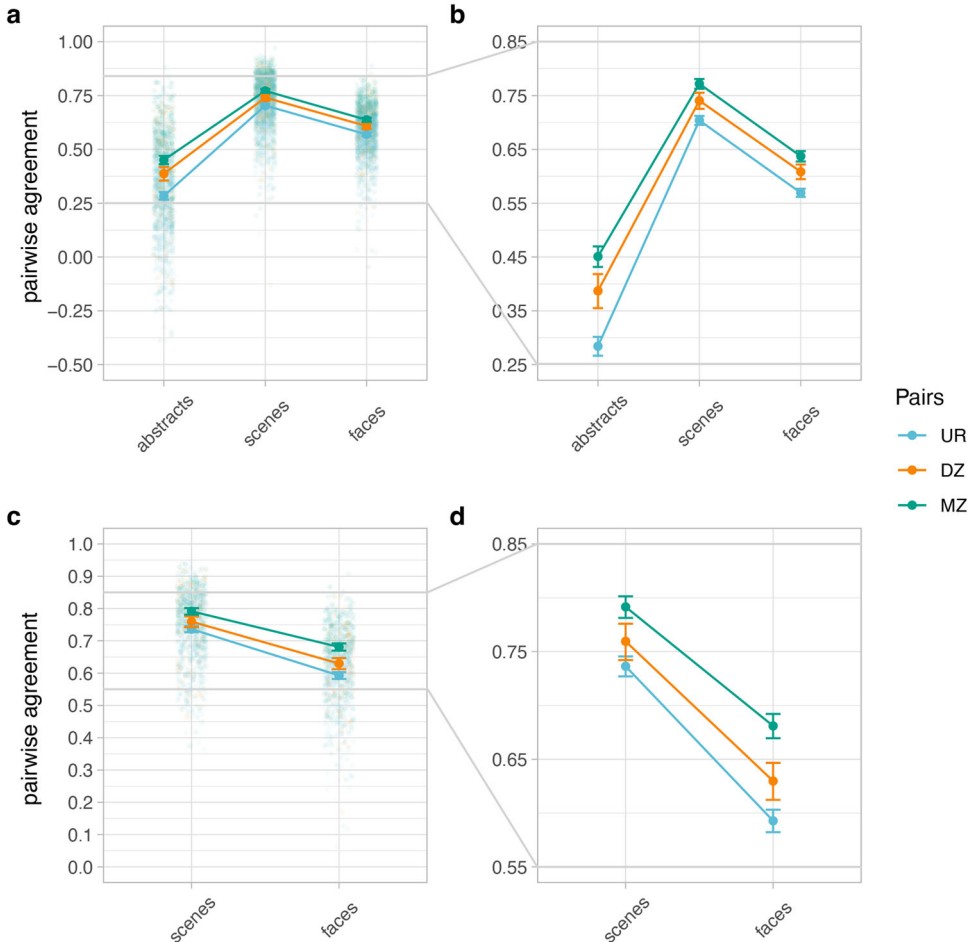

**Fig. 2 Pairwise aesthetic agreement is higher within monozygotic twin pairs.** Panels **a** and **b**, and **c** and **d** show the pairwise agreement across different visual domains and pair classes for the discovery and the validation samples, respectively. Each dot represents the pairwise agreement for one pair. **b** and **d** zoom in on the mean pairwise agreement differences between pairs. Note that MZ twins display consistently higher pairwise agreement across visual domains, albeit the effects are small. Error bars represent the 95% Confidence Intervals (95% CI). UR unrelated, DZ dizygotic, MZ monozygotic.

To frame our results within the estimated magnitude of $h^2$ for other complex human traits, we stratified our estimates within $h^2$ obtained from the largest meta-analysis of twin studies to date[50,57] (Fig. 3e). Averaged $h^2$ estimates for the major dimensions of aesthetic value for all visual domains, except taste-typicality for abstract images, ranked amongst other social and attitudinal traits yet was considerably lower than other cognitive traits for the magnitude of $h^2$. Taste-typicality for abstract images was the exception, ranking at the bottom for $h^2$ estimates, with only shared environmental but no genetic influences over its variability.

**Multivariate CTD model shows shared and distinct genetic influences on inter-individual differences in aesthetic value across visual domains.** We went on to investigate the associations between taste-typicality and evaluation-bias across the different visual domains. Correlations between taste-typicality and evaluation-bias twins scores are shown in Fig. 4. On the one hand, perhaps unsurprisingly, correlations between taste-typicality and evaluation-bias were not significant (all $p > 0.05$), except for taste-typicality for scenes and evaluation-bias for faces ($r(695) = 0.11$, $p = 0.048$), which, however, did not replicate for other twin members ($r(702) = 0.08$, $p = 0.43$, all Bonferroni corrected). This was expected, given that the two metrics strongly correlate with the two major orthogonal axes of covariation (see PCA results above). On the other hand, correlations for taste-typicality and

evaluation-bias across domains were significant, ranging from small to moderate, with the exception of taste-typicality for abstract images and faces, which was not significant, $r(688) = 0.04$, $p > 0.99$ and $r(696) = 0.09$, $p = 0.35$. Thus, individuals with taste-typicality and evaluation-bias scores in one domain tend to have similar scores in others. Results were replicated in a third, fully independent sample (Supplementary Note 9).

To partition covariation across visual domains into genetic and environmental sources, we applied the multivariate CTD (see "Methods"). This method exploits MZ and DZ cross-trait cross-twin correlations (e.g., the evaluation-bias for abstract images of twin 1 with the evaluation-bias for scenes of twin 2) to partition phenotypic covariance between traits into genetic and environmental covariance components. Further, the multivariate CTD makes it possible to quantify genetic and environmental correlations, which estimate the overlaps in variance components across traits. Multivariate models were specified following univariate results. There was a good fit for the multivariate model for taste-typicality data, which included A components for scenes and faces only, and a C component for abstract images (Fig. 5a; Saturated vs full ACE $\chi^2(\Delta df = 33) = 34.27$, $p = 0.41$, full ACE vs specified model, $\chi^2(\Delta df = 8) = 6.48$, $p = 0.59$, see "Methods"). The final model indicated that shared genetic effects accounted for more than half of the covariation between taste-typicality across visual domains (bivariate $h_b^2 = 0.66$, 95% CI [0.47, 0.85]) and that overlapping genetic effects jointly explained

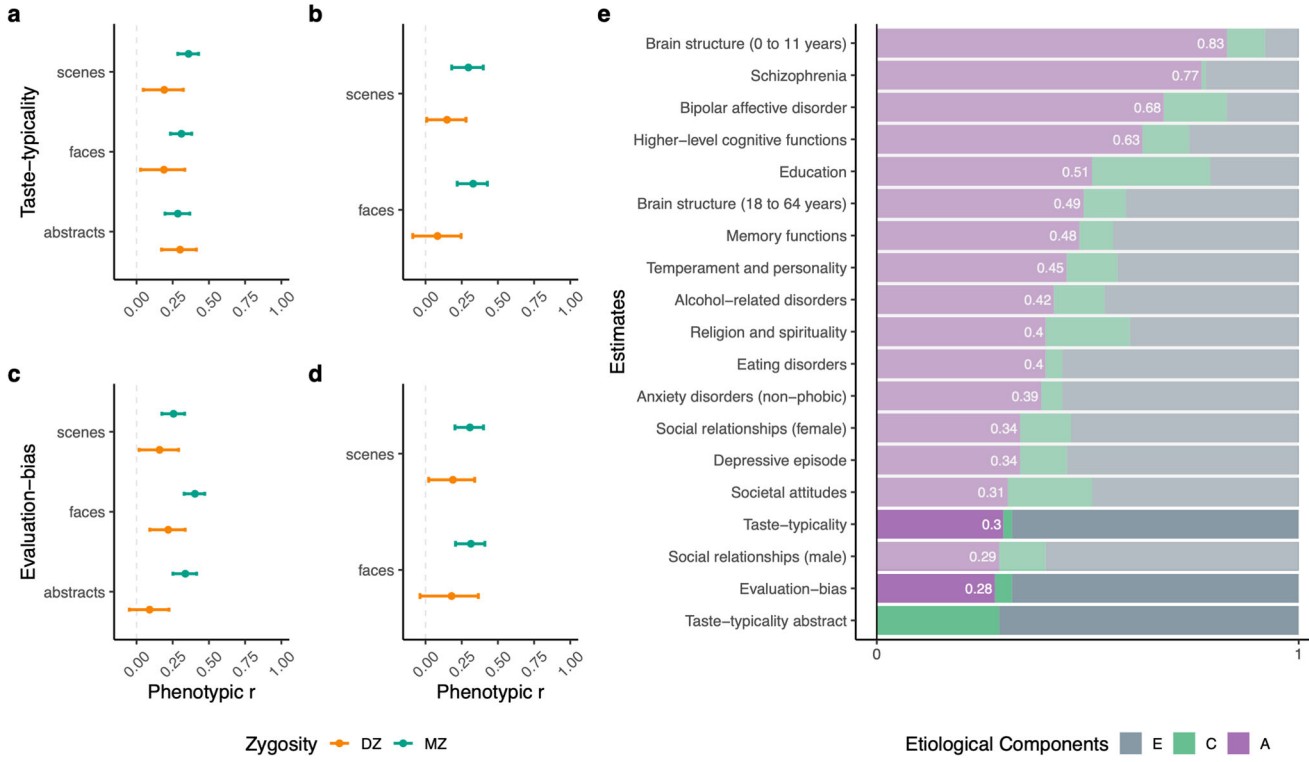

**Fig. 3 Classical Twin Design reveals etiological influences on inter-individual differences in aesthetic evaluations.** Panels **a** to **d** show the pattern of twin correlations extracted by the respective saturated models grouped by zygosity. Bars represent 95% confidence intervals (CI; see Supplementary Table 1). **e** Framing of etiological estimates for inter-individual differences in aesthetic evaluations within the etiology of other human complex traits. Bar plot with etiological environmental (E, C) and additive-genetic (A) estimates accounting for the variation of other human traits (adapted from Willoughby et al.[50] and Polderman et al.[57]). Highlighted: Averaged $h^2$, shared, and unique environmental estimates for taste-typicality for scenes, and faces, and evaluation-bias for all domains, and shared and unique environmental estimates for taste-typicality for abstract images. A additive genetic, C common environment, E unique environment and unsystematic effect (e.g., measurement error).

variation in taste-typicality for images of scenes and faces (genetic correlation ($\rho A$), $\rho A = 0.53$, 95% CI [0.37, 0.69]). This pattern was in contrast with overlapping unique environmental effects on inter-individual differences for taste-typicality, which were small across all visual domains.

Similarly, the AE multivariate model fit for evaluation-bias data was good (Fig. 5b; Saturated vs full ACE $\chi^2(\Delta df = 33) = 40.22$, $p = 0.18$, full ACE vs specified model, $\chi^2(\Delta df = 6) = 1.77$, $p = 0.94$). Bivariate heritability estimates spanned from bivariate $h_b^2 = 0.31$, 95% CI [0.2, 0.42], between abstract images and scenes, to bivariate $h_b^2 = 0.44$, 95% CI [0.31, 0.56], between abstract images and faces. Genetic correlations were moderate, ranging from $\rho A = 0.51$, 95% CI [0.36, 0.65], between scenes and faces, to $\rho A = 0.59$, 95% CI [0.43, 0.72], between abstract images and scenes, respectively. Unlike for taste-typicality, we also found environmental correlations to be substantial (Fig. 5c).

Findings from the validation sample (Fig. 5d) were consistent with the results from the discovery sample reported above. In particular, the multivariate AE model for the taste-typicality data for scenes and faces, specified in the discovery sample, resulted in a good fit for the data in the validation sample (Saturated vs full ACE $\chi^2(\Delta df = 17) = 26.39$, $p = 0.07$, full ACE vs specified model, $\chi^2(\Delta df = 3) = 0.86$, $p = 0.83$). Similarly, an AE multivariate model was a good fit for the evaluation-bias data in the validation sample (Saturated vs full ACE $\chi^2(\Delta df = 17) = 13.25$, $p = 0.72$, full ACE vs specified model, $\chi^2(\Delta df = 3) = 0.22$, $p = 0.97$).

**Sensitivity analyses discount contributions of potential confounding effects.** To account for possible confounds, we used

non-aesthetic ratings obtained in the validation sample on a control task (see "Methods") using the same face stimuli[45]. We calculated control-typicality and control-bias scores as the *mm2* and the within-individual rating average of the control task. Control-typicality reflected the tendency to display typical scores on a non-aesthetic image rating task. Control-bias reflected the general tendency to give high or low ratings on average on a non-aesthetic task. Taste-typicality and control-typicality, and evaluation-bias and control-bias scores were positively correlated (Supplementary Fig. 8), indicating possible confounding effects. To assess whether such effects biased our etiological estimates, we regressed taste-typicality and evaluation-bias on the control-typicality and control-bias scores (in addition to sex and age) and re-ran both univariate and multivariate CTD on the residuals. The correlations between MZ and DZ with residualised taste-typicality and evaluation-bias were similar to those extracted by the model fitted without accounting for possible confounders (Supplementary Table 1). Crucially, univariate and multivariate CTD modelling indicated that the same models specified for uncontrolled scores fitted residual scores with estimates for the genetic components being similar to those obtained without controlling for confounders, with $h^2$ being equal to $h^2 = 0.28$, 95% CI [0.18, 0.38] and $h^2 = 0.29$, 95% CI [0.18, 0.39], for residualised taste-typicality, and $h^2 = 0.27$, 95% CI [0.17, 0.37] and $h^2 = 0.29$, 95% CI [0.19, 0.39], for residualised evaluation-bias, for scenes and faces, respectively (see Supplementary Table 2 and Supplementary Fig. 9 for further $h_b^2$, $\rho A$ and $\rho E$ estimates). Therefore, sources of variability in major dimensions of aesthetic value, as found in this study, are not confounded by individuals' general typicality or overall biases in rating scale uses.

**Table 2 Univariate modelling of genetic and environmental contributions to inter-individual differences in aesthetic evaluations.**

| Images | Sample | Model | −2LL, AIC | df, $\chi^2$ | A | C\|D | E |
|---|---|---|---|---|---|---|---|
| **Taste-typicality** | | | | | | | |
| Abstract Images | Discovery | CE* | 349.52, 359.52 | 1409, 0.04 | 0 | 0.29 [0.22; 0.36] | 0.71 [0.64; 0.78] |
| | | ACE | 349.47, 361.47 | 1408, 8.41 | −0.03 [−0.31; 0.37] | 0.32 [0.05; 0.55] | 0.71 [0.63; 0.80] |
| Scenes | Discovery | AE* | 373.23, 383.23 | 1518, 0.03 | 0.36 [0.29; 0.43] | 0 | 0.64 [0.57; 0.71] |
| | | ACE | 373.20, 385.20 | 1517, 4.65 | 0.34 [0.04; 0.66] | 0.02 [−0.27; 0.29] | 0.64 [0.57; 0.71] |
| Scenes | Validation | AE* | 276.70, 286.50 | 1204, 1.76 | 0.30 [0.19; 0.39] | 0 | 0.70 [0.61; 0.81] |
| | | ACE | 276.50, 288.50 | 1203, 12.45 | 0.29 [−0.05; 0.64] | 0.00 [−0.30; 0.28] | 0.70 [0.60; 0.82] |
| Faces | Discovery | AE* | −804.66, −795.66 | 1505, 0.05 | 0.33 [0.25; 0.40] | 0 | 0.67 [0.62; 0.76] |
| | | ACE | −804.71, −792.71 | 1504, 6.40 | 0.29 [−0.02; 0.63] | 0.03 [−0.29; 0.32] | 0.67 [0.60; 0.75] |
| Faces | Validation | AE* | −439.82, −429.82 | 1161, 0.84 | 0.31 [0.21; 0.41] | 0 | 0.69 [0.59; 0.79] |
| | | ADE | −440.65, −428.65 | 1160, 7.81 | 0.00 [−0.69; 0.66] | 0.33 [−0.35; 1.04] | 0.67 [0.57; 0.78] |
| **Evaluation-bias** | | | | | | | |
| Abstract Images | Discovery | AE* | 3523.48, 3533.48 | 1410,1.17 | 0.32 [0.24; 0.40] | 0 | 0.68 [0.60; 0.76] |
| | | ADE | 3522.31, 3534.31 | 1409,2.74 | 0.02 [−0.54; 0.56] | 0.31 [−0.24; 0.89] | 0.68 [0.58; 0.75] |
| Scenes | Discovery | AE* | 3110.07, 3120.07 | 1520,0.18 | 0.26 [0.18; 0.33] | 0 | 0.74 [0.67; 0.82] |
| | | ACE | 3109.88, 3121.88 | 1519,10.56 | 0.19 [−0.11; 0.51] | 0.06 [−0.23; 0.34] | 0.74 [0.67; 0.82] |
| Scenes | Validation | AE* | 2855.31, 2865.31 | 1203,0.21 | 0.31 [0.21; 0.40] | 0 | 0.69 [0.60; 0.79] |
| | | ACE | 2855.13, 2867.13 | 1202,6.85 | 0.24 [−0.13; 0.62] | 0.07 [−0.28; 0.38] | 0.69 [0.60; 0.80] |
| Faces | Discovery | AE* | 3062.91, 3072.91 | 1507,0.07 | 0.41 [0.33; 0.47] | 0 | 0.59 [0.53; 0.66] |
| | | ACE | 3062.84, 3074.84 | 1506,7.43 | 0.37 [0.10; 0.66] | 0.03 [−0.23; 0.27] | 0.60 [0.53; 0.67] |
| Faces | Validation | AE* | 3164.23, 3174.23 | 1165,0.05 | 0.32 [0.21; 0.41] | 0 | 0.68 [0.59; 0.79] |
| | | ACE | 3164.19, 3176.19 | 1164,2.64 | 0.27 [−0.16; 0.74] | 0.05 [−0.40; 0.43] | 0.69 [0.59; 0.79] |

Sex and age are included as covariates in every model. Significance and comparative statistics for the reduced models (AE and CE) are obtained by comparison with the respective full ACE|ADE model. (All ACE|ADE models fit were considered good compared to the respective full saturated models). Reported $\chi^2$ df = 1 for the full vs. reduced model (e.g., AE vs. ACE) and df = 6 for all saturated vs. full model. * Final reduced models.
−2LL − 2 Log likelihood, AIC Akaike information criterion, df degrees of freedom, A additive genetic, C|D common environmental or dominance genetic, E unique environmental and error.

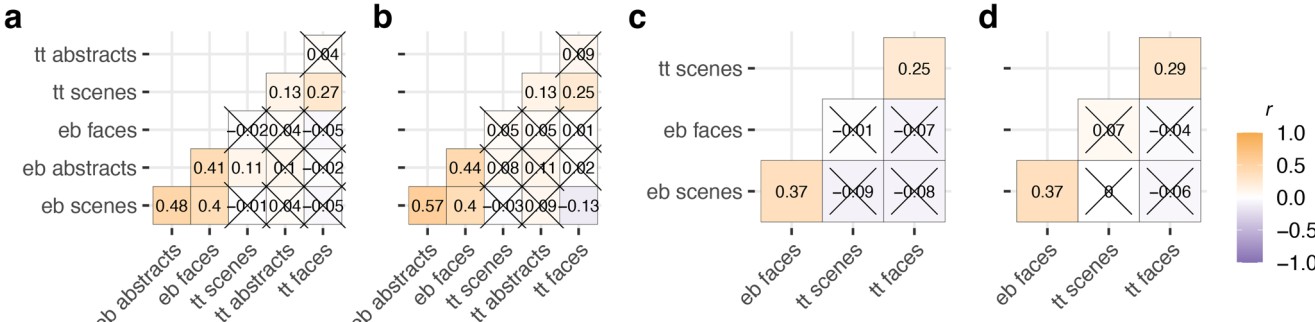

**Fig. 4 Phenotypic correlations for taste-typicality (tt) and evaluation-bias (eb) across visual domains.** Pearson phenotypic correlations (*r*) for the taste-typicality and evaluation-bias were obtained separately from pairs' members to avoid familial confounding effects. **a** and **b** discovery and **c** and **d** validation samples, respectively. **a** and **c** depict correlations computed on only one twin per pair; **b** and **d** depict correlations computed from the other twins. Crosses represent the non-significant Pearson correlations (*p* < 0.05; Bonferroni corrected). tt taste-typicality, eb evaluation-bias.

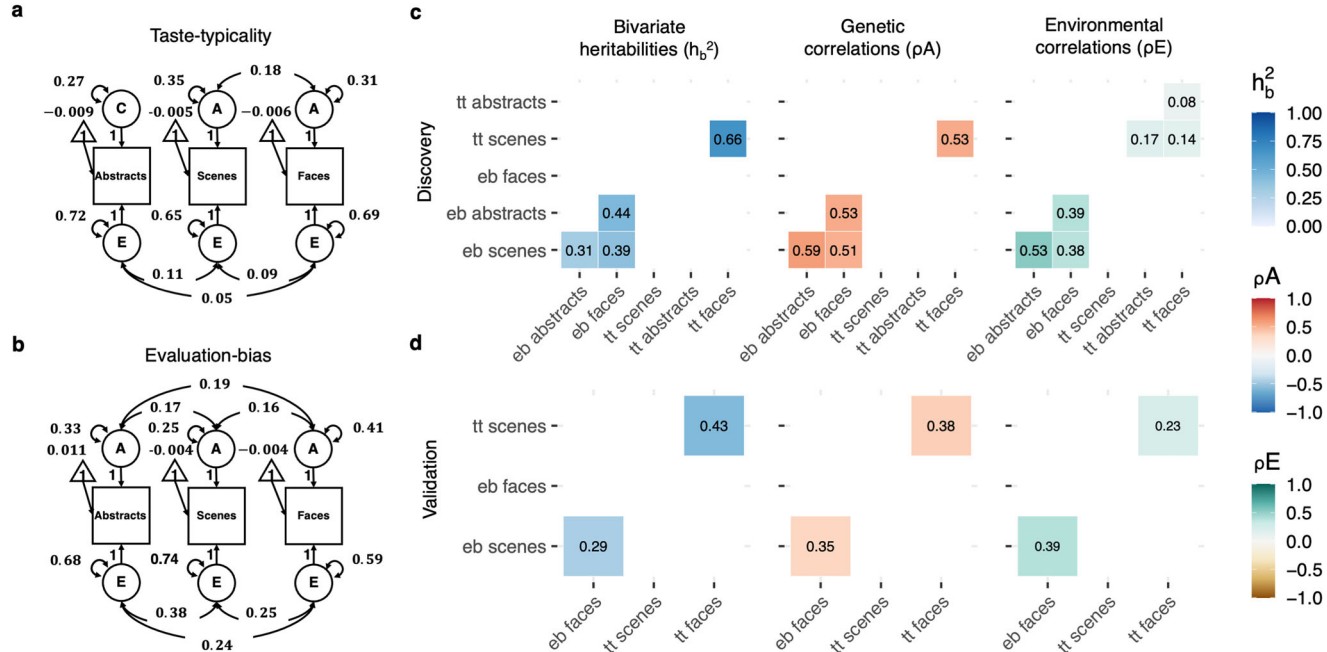

**Fig. 5 Multivariate modelling results indicate partially shared genetic effects across visual domains. a** Simplified path diagrams for the specified multivariate SEM informed by CTD fit on MZ and DZ twin discovery sample taste-typicality standardised data. Triangles represent means; squares represent observed variables; circles represent latent components; double-headed arrows starting and ending in the same square or circle represent variances; double-headed arrows starting on one circle and ending in another circle represent covariances; one-headed arrows represent paths. Example: the variance for the additive genetic component for the taste-typicality score for scenes is $\sigma^2_{Asce} = 0.35$, while the additive genetic covariance between taste-typicality for scenes and faces is $\sigma_{AsceAfac} = 0.18$. **b** The multivariate SEM for the evaluation-bias data. **c** Multivariate CTD results, from left to right: Bivariate heritability, $h^2_b$ (e.g., $h^2_{b\_scefac} = \sigma_{AsceAfac}/(\sigma_{AsceAfac}+\sigma_{EsceEfac})$ $0.18/(0.18 + 0.09) = 0.67 \sim 0.66$ in panel **b**; difference is due to rounding), i.e., the amount of covariance between two traits accounted for by additive genetic variance (the amount accounted for by environmental components is simply $1–h^2_b$); and the genetic and environmental correlations matrices, (e.g., $\rho A\_{scefac} = \sigma_{AsceAfac}/\sqrt{(\sigma_{Asce}^2*\sigma_{Afac}^2)} = 0.18/\sqrt{(0.35*0.31)} = 0.55 \sim 0.53$; difference is due to rounding; and $\rho E\_{scefac} = \sigma_{EsceEfac}/\sqrt{(\sigma_{Esce}^2*\sigma_{Efac}^2)}$). **d** Matrices obtained from the validation sample[45].

## Discussion

Building on previous work by Germine et al.[32] and Sutherland et al.[45], we confirmed that the environment shapes inter-individual differences in the aesthetic evaluations of visual images, yet revealed that genetics, too, play an important role. On the one hand, we found that genetic influences have small effects on which visual images people prefer, as genetically related individuals tend to display only slightly more similar ratings of images than unrelated individuals, regardless of the evaluated visual domain. These results strengthen earlier claims that environmental influences are a source of variation in aesthetic preferences for faces[32,45] and extend such claims to previously not investigated visual domains such as abstract images and scenes. On the other hand, we show that variability in taste-typicality (how similar individuals' preferences are to the group preferences) and evaluation-bias (the overall aesthetic value evoked by visual images) are substantially influenced by genetic effects. In contrast to the aesthetic agreement between individuals (the pairwise similarity between individuals' preferences), these two metrics represent more foundational measures of aesthetic evaluations and how they systematically vary inter-individually. These latter results challenge the traditional view that environmental experiences alone shape inter-individual differences in the formation of aesthetic value across various visual domains.

Regarding taste-typicality for faces, perhaps unsurprisingly, our results are broadly in line with the claims of Germine et al.[32] and Sutherland et al.[45]. Taste-typicality is highly similar to the original metric employed by the authors. As such, the resulting heritability for taste-typicality for faces should not be considered entirely novel, despite some notable differences in analytic procedure.

However, our finding partially contrasts the previous authors' conclusion by challenging the extent to which the originally employed metric can be used to infer individual preferences for, or impressions of, faces. For those studies, comparisons between MZ and DZ twin individual preference scores were used to quantify the extent to which genetic or environmental factors contribute to variation in individual preferences or impressions. Yet, we found that aesthetic agreement and taste-typicality are not equivalent measures for individual preferences. Our work indicates that metrics that quantify how typical individuals' aesthetic evaluations are compared to the group average hold only a small amount of information about the etiology of actual preferences[45] within a pair, for any domain. From an analytical point of view, as extensively discussed in the supplement, this stems from the observation that two individuals might display similar taste-typicality scores and yet strongly disagree on their aesthetic preferences. This led us to conclude that comparison between MZ and DZ twin taste-typicality scores can be used to obtain descriptions of the etiology of how typical individuals' aesthetic evaluations are, such as those described by Chen et al.[13], yet these scores are not particularly suited to studying pair preferences.

Moreover, we speculate that the differences between aesthetic agreement and taste-typicality could also be, in part, reconceptualised as reflecting individual differences in arbitrary specific states versus more general systematic traits. From both theoretical and empirical work, we know that aesthetic preferences can be context dependent[60,61] and vary over time[62]. Yet, we also know that individuals vary systematically in the way they aesthetically evaluate sensory stimuli[13]. Within these frameworks, aesthetic

agreement and hence aesthetic preferences for a set of sampled images, would thus partially represent a shared state of preferences between pairs of individuals, while taste-typicality (as well as evaluation bias) would represent a more stable individual trait, less bounded by the actual sample of images and evoked states. This view aligns with previous work suggesting taste-typicality systematically correlates across sensory modalities[13] and with the current moderate correlation within sensory modality across different visual domains. Moreover, under such a framework, we would expect aesthetic agreement to show lower repeatability over larger periods of time than taste-typicality, a trend that we start to observe in the third independent and unrelated sample, as further discussed in the supplement. Nevertheless, we note that more research is needed to test whether this state versus trait reconceptualisation is consistent with aesthetic agreement versus taste-typicality (and evaluation-bias) repeatability over a large period of time.

Another interesting point pertains to the observation of a strong correlation between evaluation-bias and the first axis of inter-individual variability in aesthetic value, with taste-typicality being strongly correlated only with the second axis. These findings contrast a recent study that found taste-typicality rather correlated with the first but not the second axis[13]. Our results indicate that evaluation-bias systematically captures the majority of variance in aesthetic valuations of different visual domains. Furthermore, strengthened by the moderate estimates for heritability and genetic correlations across visual domains, our findings suggest that evaluation-bias is a meaningful foundational metric of inter-individual differences in the way visual sensory stimuli acquire aesthetic value[33]. This conclusion is further supported by our sensitivity analysis. Similar to what others have done for individual preferences[32], and what we have also done for taste-typicality, we correct evaluation-bias for overall scale bias uses, obtained in a non-aesthetic task, ensuring the specificity of the metrics used to assess variability in aesthetic evaluation. This step removes the influence of psychological traits that relate to a general tendency to display high or low average ratings, as well as to display ratings similar to the mean, regarding the task. However, we also acknowledge that our residualisation, or any residualisation in general[63], cannot remove all possible confounders. It is, therefore, possible that some of the $h^2$ estimated in this study is partially shared with the well-known heritability of general traits such as personality, especially the (moderately heritable) part of openness-to-experience related to aesthetic processes[43].

Taste-typicality and evaluation-bias heritability ($h^2$), as estimated in this study, have the potential to inform both evolutionary and comparative aesthetics studies. It is well known that $h^2$ constrains response selection of any trait over subsequent generations[52]. Specifically, an $h^2$ near zero implies little to no room for response selection and, hence, no room for evolutionary pressures to act upon a quantitative trait within a population and an environment. Within this context, our results suggest no room for genetically mediated evolution of taste-typicality of abstract images, while some room for taste-typicality for faces and scenes and evaluation-bias across all visual domains. Further studies in humans could extend these results across diverse populations and environments and to the formation of the aesthetic value of other man-made objects, such as art objects, answering whether changes in aesthetic value for artefacts are purely due to cultural evolutionary pressures and/or drift-like phenomena[64], or whether genetic constraints exist for art-like objects too. Furthermore, comparative studies could adapt similar unidimensional metrics to the one employed in this study (e.g., taste-typicality) to allow the study of problematic multidimensional measures of preferences in non-human animals. This could be informative for addressing open questions in the field of aesthetics[21], such as whether it is possible to study inter-individual differences in aesthetic value in non-human animals.

Beyond $h^2$ estimates, we also found that shared environmental effects (C) on inter-individual differences in fundamental metrics of aesthetic value, if present, are small to negligible, except for taste-typicality for abstract images. It is known that mere exposure to[12], associations with[35], and self-relevance of[65] visual objects correlates with their perceived aesthetic value. As such, given that twins are exposed to similar environments and thus are expected to be exposed to more similar visual sensory stimulation over a lifetime, the finding that models with purely additive genetic and unique environmental sources of differences can mostly explain variability in taste-typicality and evaluation-bias might seem counter-intuitive. Yet this finding falls well under the second law of behavioural genetics, which states that the "effect of being raised in the same family is smaller than the effect of genes"[66], at least for within-group variability, and the empirical support for it over many other quantitative traits[57]. Moreover, it is also possible that as for other cognitive traits, such as general intelligence, shared environmental effects over aesthetic value formation matter most while the environment is shared and within certain developmental windows, with a gradual decrease into adulthood (generally known as the Wilson effect[50,67]). The observation that C is trivial in the adult samples analysed in this study would be consistent with the latter hypothesis. Further studies with younger individuals would be needed to formally test whether family non-genetic effects on aesthetic value formation indeed vary over the lifespan (however, see Supplementary Note 10 and Supplementary Figs. 10 and 11 for a detailed discussion on statistical power to detect C).

Nonetheless, we found evidence of shared environmental influences on variation in taste-typically for abstract images, representing a clear violation of the second law of behavioural genetics. This exceptional finding bears on the important question of how aesthetic processes may differ for biological (natural) versus artefactual (non-natural) categories of objects (see refs. [11,68]) and how effects within families can influence individuals differently across domains. We note that, as previously suggested by the literature, individuals tend to show a high degree of aesthetic agreement for images of faces and natural landscapes, both of which are natural kinds, but much less for abstract images[11,31,69,70]. Our finding of a behavioural trait in an adult sample in which individual differences appear to be purely influenced by environmental factors, both shared and unique (i.e., taste-typicality for abstract images), raises the question of whether aesthetic processes underlying the evaluation of other human-made artefactual objects, such as visual art and music, are free from genetic predispositions. Recently, it has been proposed that inferential processes about the sensory world, both perceptual and aesthetic, can be understood as falling between two extremes; one (inferential processes about natural kinds) being constrained by genetic predispositions, while the other (inferential processes about non-natural kinds) is mainly shaped by environmental exposure[68]. Under this framework, aesthetic value derived from natural images is more constrained by a priori predispositions than the value that is derived from images belonging outside natural categories. Our findings may provide partial support for this hypothesis, in that there was a robust genetic correlation between taste-typicality for scenes and faces, but an absence of evidence for genetic effects on variation in taste-typicality for abstract images. However, in addition to previous literature[32,71] indicating that variability in perceptual processes, more than aesthetic processes, correlates mostly with genetic differences amongst individuals, the magnitudes of effects in our results indicate smaller genetic constraints than those originally hypothesised to influence variation in the aesthetic evaluation of

natural kinds. If such constraints do exist and are captured by the genetic components found in this study, they contribute to less than half the variability in the major axis of inter-individual variation in aesthetic value. This leaves unsystematic and environmental sources (rather than systematic genetic ones) as the leading source of differences in aesthetic evaluation, even for images that belong to natural categories.

Beyond univariate $h^2$ estimates, we also found that the genetic effects contribute to the major dimensions of aesthetic value extending across visual domains. There were substantial genetic overlaps for taste-typicality and evaluation-bias for abstract images, scenes and faces. This was indexed by the amount of covariance explained by genetic factors (bivariate heritability estimates) between taste-typicality for scenes and faces and evaluation-bias for all visual domains, and by the shared genetic effects on the variation of such traits across visual domains (genetic correlations). The first finding indicates that genetic influences partially account for the observed relationship across domains for both taste-typicality and evaluation-bias. The second finding shows that the genetic effects on such aesthetic dimensions overlap across domains, suggesting that shared genetic factors underlie variation in major dimensions of visual aesthetic value. Nonetheless, genetic correlation estimates across domains were far below 1, indicating that different genetic effects also play a role.

We note that our study has some limitations. In re-analysing existing datasets to shed light on some questions, we were necessarily confined to the image domains and measurement techniques used in the original studies. For example, the sets of scenes contained images such as photographs of construction sites or classrooms, which diminish the extent to which we could interpret our results considering a natural versus artefactual kind distinction. Further, ratings collected differed slightly between the two samples, with differences in the numerical extremes of the possible answers (e.g., from 1-to-7[32] to 1-to-9[45]). Furthermore, given the nature of our re-analysis, the extent of overlap between the two samples is not known. Moreover, we assumed that residual differences between the repeated ratings were a type of measurement error, an implicit assumption that does not align with findings from the empirical aesthetics literature on the effect of repeated exposures (e.g., systematic effects of "mere exposure"[12] or habituation[72]).

Despite these limitations, the results from the multilevel modelling demonstrated that different domains captured a sufficiently diversified amount of idiosyncrasies, consistent with growing evidence indicating that common aesthetic preferences emerge from shared representations of sensory experiences[11,31]. They also revealed highly replicable variance component estimates across the samples, suggesting that slight differences in the numerical extremes did not substantially affect the level of inter-individual differences in response. Moreover, we note that, although the extent of overlap between the two samples is not known, two random samples of size 1547 and 1231 (i.e., the sample size of the Germine et al. and Sutherland et al. studies) drawn from a sample bigger than 40,000 (i.e., the estimated sample size of the Australian Twin Registry[73]) would, on average, result in only a sample overlap of less than 50 individuals (see Supplementary Note 11 and Supplementary Fig. 12). Furthermore, given that the analysis in the replication sample aimed to show the robustness of the results to possible confounders and ensure that estimates were relatively stable, we believe that the likely inclusion of the same individuals in the discovery and validation samples would have no impact on our conclusions, even assuming a more extensive sample overlap. Finally, these results revealed a minimal systematic effect of two image repetitions on overall variance in aesthetic ratings, suggesting that

repeated exposure impacted ratings negligibly, justifying our assumption that, in these samples, any such variation is largely measurement noise.

In sum, we found monozygotic (MZ) twins to be more similar than dizygotic (DZ) twins in their aesthetic evaluations of abstract images and scenes, and of images of faces. Differences were small, suggesting weak genetic effects on visual aesthetic preferences. Nonetheless, we found that genetic factors accounted for about one-third of the variance in how similar individuals' preferences are to group preferences (taste-typicality) and in the overall aesthetic value evoked by visual images (evaluation-bias), two major dimensions underlying inter-individual differences in aesthetic valuation. Findings were robust to replication and confound. Compared with other complex human traits, these estimates put variation in aesthetic evaluations on par with other social and attitudinal traits[50] for the amount of variance explained by genetic factors. Yet, we also found the presence of shared and unique environmental effects for inter-individual differences in taste-typicality for abstract images. These particular results align with the view that, for some traits, environmental exposure alone influences the formation of aesthetic value, albeit only for one dimension of aesthetic value (taste-typicality) and one visual domain (abstract images), raising interesting questions about the formation of aesthetic value for other human-made artefactual objects. Finally, genetic effects explained a substantial part of the association between taste-typicality for images of scenes and faces, and evaluation-bias for all visual images in both samples investigated in this study. Genetic correlations suggest a partial overlap between such genetic sources of variation, indicating both shared and distinct genetic contributions to variation in aesthetic value, especially between taste-typicality for images of scenes and faces.

Our analyses show that, while the environment is indeed the leading source of differences in visual aesthetic evaluations, it is not the only influence. Although aesthetic evaluations of visual images represent a fundamental, inherently subjective, and pervasive aspect of internal representations of the external sensory world, they are also partially influenced by the genetic variants that people carry.

## Methods

**Stimuli**. The set of stimuli used by Germine et al.[32] was originally divided by the authors into three groups: abstract images, scenes and faces. A total of 50 abstract images and 50 scenes were collected from two previous studies[31,74]. Abstract images were composed of six sub-domains: "(1) abstract shapes created using Maya 3D rendering software, (2) kaleidoscopic images constructed by reflecting a sliver of a real-world image about a number of symmetry axes, (3) pseudo-coloured electron microscope images obtained from the Centre for Microscopy and Microanalysis at The University of Queensland, (4) fractal images created with publicly available interactive programs, (5) satellite imagery courtesy of the US Geological Survey, and 6) the other category of images collected from public Internet sources"[31, p.4]. Images of scenes were originally selected by Vessel et al.[31]. For faces, 200 images of faces were selected from four different resources. Respectively, (1) 50 from the MIT database, (2) 50 from the Glasgow Unfamiliar Face Database, (3) 50 from the GenHead software, and (4) 50 as a collection of various sources (Facial Recognition Technology Database, the NimStim Set of Facial Expressions, and the Karolinska Directed Emotional Faces; see Germine et al.[32] for details). The final sets of images were divided into six blocks: one block for abstract images (50 images + 15 repeated images), one block for scenes (50 images + 15 repeated images) and four blocks for images of faces (50

images + 15 repeated images per block). Participants were not made aware of such subdivisions by the original authors. Repeated images were added to each block by Germine et al.[32] to assess the intra-individual reliability ($R_{intra\text{-}xx}$) of the ratings. Participants were asked to rate how attractive (or visually pleasing, with the latter as an addition to the former for images of scenes and faces) images were using a 1-to-7 rating scale.

The set of stimuli used by Sutherland et al.[45] was originally divided into two groups, sceneries, here referred to as scenes, and faces. For scenes, 50 images were selected following the same criteria used by Germine et al. from the same resource. For faces, 100 images of faces were selected from the 10 K face database[75], which was chosen as the only resource to maximise the ecological validity of the stimuli. The final images were divided into two blocks: one block for scenes (50 images + 24 repeated images) and one for faces (100 images + 50 repeated images per block, see ref. [45] for details). Participants were asked to rate images using a 1-to-9 rating scale.

**Exclusion criteria.** Similarly to Germine et al.[32] and Sutherland et al.[45], we excluded participants with ratings $sd = 0$, within each visual domain. Additionally, similarly to Vessel et al.[11] and Chen et al.[13], we excluded $R_{intra\text{-}xx} < 0.5$ individuals within each visual domain, indicating poor test-retest reliability in ratings. $R_{intra\text{-}xx}$ was computed within each participant as the Pearson correlation of their rating for the repeated images per visual domain. Additionally, given the sensitivity to extreme outliers in many of the statistical tests we conducted later, we excluded pairs and individuals with metrics with values above 3 times the Inter Quartile Range (IQR) in any metrics per visual domain from further respective analyses.

**Pairwise aesthetic agreement.** The pairwise aesthetic agreement was quantified as the inter-individual Pearson correlation ($r_{inter}$) of pairs' ratings. To create a familial unrelated reference for the pairwise aesthetic agreement, we additionally computationally created unrelated pairs (UR) with pseudo-random pairing by matching every second twin member of a pair with a first member from another pair. UR pairs were matched for sex and did not differ in age within each domain (Supplementary Note 6). To conduct statistical analysis on the pairwise aesthetic agreement scores, we transformed $r_{inter}$ to $z_{inter}$ following Fisher's z transformation. Similar to Vessel et al.[11], we reported r values transformed back from the z values to aid the interpretability of the results.

**Taste-typicality and evaluation-bias.** The mean minus 2 (*mm2*), adapted from the mean minus one (*mm1*; Vessel et al.[11]) to account for familial resemblances, was computed to quantify taste-typicality scores[13]. Taste-typicality reflects the similarity between each individual's set of aesthetic ratings per visual domain and the group's average ratings. The *mm2* is simply the Pearson correlation between the ratings given by one participant with the average ratings of every other participant, excluding the participant's twin (hence mean minus two), and is calculated independently for each visual domain. We used Fisher z transformation to transform the *mm2* values to further carry computations on taste-typicality scores (*mm2* values reported in the result section were z-to-r transformed to aid interpretability, as in Vessel et al.[11]). It is important to note that our taste-typicality score is almost equivalent to the individual preferences score of Germine et al. (except for accounting for familial resemblances and not being regressed on other aesthetic ratings[32]) and to the taste-typicality score of Chen et al. (except for accounting for familial resemblances and being calculated from unstandardised

data[13]). Evaluation-bias was measured as the individual average of aesthetic ratings per visual domain, reflecting the magnitude of the overall pleasantness experienced by individuals when evaluating visual images.

### Statistics and reproducibility

*Variance partitioning coefficients.* To calculate the amount of variance in visual aesthetic value shared and unique across individuals, we carried out Variance Component Analysis (VCA[56]) through random intercept-only multilevel models[45,76]. Multilevel models were independently fitted on each of the three visual domains using the following formula:

$$Lme4 :: lmer(Rating \sim 1 + ((1|Individual) + (1|Image)$$
$$+ (1|Individual : Image) + (1|Exposure)$$
$$+ (1|Exposure : Individual) + (1|Exposure : Image)))$$

Similarly to Sutherland et al., the random effects of the models included the image and the observer (i.e., the individual) intercepts and the interaction between the two. Additionally, as in Martinez et al.[56], our model included the exposure effect for images rated more than once and its interaction with the image and the observer effects. This enabled us to partition the percentage of variance in visual aesthetic evaluations explained by the image (shared across observers), the observer, and the interaction between the two (unique to the observers). This was important as it allowed us to quantify idiosyncrasies within a domain: the interaction between the individual evaluating and the stimulus evaluated, from the overall evaluation within a domain. For example, two participants might interact with abstract images very similarly (e.g., give ratings of 1, 2, 3, and 5, 6, 7 to the same three images, respectively) but express a very different overall rating for images in general (e.g., 2 and 6 respectively). Further, it allowed us to estimate the amount of systematic variance explained by the interaction between the observer and the image with the exposure (see Supplementary Note 3 for details). This further allowed us to justify the averaging procedure within participants' repeated ratings to increase the signal-to-noise ratio. To avoid familial confounding effects, we ran two multilevel models per domain, one for each twin of a pair. Results are shown for the first pair's members alone (see Supplementary Fig. 3 for models fitted on the second twin members). Variance Partitioning Coefficients (VPC) were computed as the amount of variance explained by one effect (e.g., ind) over the total variance, e.g.:

$$VPC_{ind} = \frac{\sigma^2_{ind}}{\sigma^2_{img} + \sigma^2_{ind} + \sigma^2_{img*ind} + \sigma^2_{exp} + \sigma^2_{ind*exp} + \sigma^2_{img*exp} + \sigma^2_{res}}$$

Where $\sigma^2$ is the variance of the individual (ind), image (img), exposure (exp), terms and their interaction and the residual (res) one.

*Principal component analysis.* To reduce the dimensionality of the aesthetic rating data, we conducted Principal Component Analysis (PCA) independently per each visual domain, where each axis represents an image, each coordinate an individual rating for that image, and each individual occupies a specific point in space. PCA is a data dimensionality unsupervised learning procedure, which geometrically projects data to the sequentially orthogonal axis (PC; principal components) for which the projected data have their variance maximised[77]. PCs were obtained through singular value decomposition of the unscaled and centred data matrix, using the base R function prcomp(). To compare the extent to which taste-typicality and evaluation-bias related to the major axes of variation in ratings, we computed Pearson correlations between the individual evaluation-bias and the taste-

typicality scores and the first and second PC scores, respectively. The correlations between taste-typicality and the second PC scores were conducted after Fisher's z transformation of the taste-typicality values[78].

*Pairwise aesthetic agreement analysis.* We conducted statistical analysis on the z-transformed value ($z_{inter}$) of the pairwise agreement, quantified as a pair's inter-individual correlation ($r_{inter}$), following Fisher's z transformation for distribution of $r$ values[78]. Means were calculated from the z distribution. Similar to Vessel et al.[11], to aid the interpretability of the results, we reported $r$ values transformed back from the z values. We extended our analysis beyond taste-typicality scores[13,32,45], as an individual with similar individual preferences (see Supplementary Note 4 for details) can have very different aesthetic ratings, opting instead for $r_{inter}$ as a proxy for pairwise agreement, with the additional step of averaging repeated image ratings before computing correlations to increase the signal to noise ratio of participants' ratings. To compare the differences in $z_{inter}$, a $3 \times 3$ type III ANOVA was performed using the ANOVA function as follows:

$$ANOVA(lm(z_{inter} \sim Pair * Domain, Data), contrasts$$
$$= contrasts\_list, type = 3)$$

We performed pairwise comparisons on the estimated marginal means using the emmeans package[79]. Family-wise tests were Bonferroni corrected for multiple comparisons (i.e., 3 for pairwise domain contrasts, 3 for pairwise pair class contrast, and 9 for pairwise pair class | domain contrast). A graphic visualisation of the analysis can be found in Supplementary Figs. 6 and 7.

*Univariate modelling.* Variance can be partitioned into either genetic or environmental sources. Whereas genetic sources of variation are considered to be either additive (additive, A) or non-additive (dominance, D), environmental sources of variation are considered to be shared (common, C) or unique to an individual (environment, E), with the latter also including measurement errors[57]. We performed model fitting using OpenMx on taste-typicality and evaluation-bias for abstracts, scenes and face images independently. Univariate models were adapted from Hermine Maes, available at https://hermine-maes.squarespace.com/. Assumptions (mean similarity and homogeneity of variance across twin order and zygosity) were tested in a saturated model. The goodness of the fit was evaluated by assessing Akaike's Information Criterion (AIC; lowest AIC meaning more parsimonious model) and by assessing the p-value associated with the $\chi^2$ value (alpha = 0.05; p-values higher than 0.05 do not indicate significant deterioration of the model fit). The $\chi^2$ test was performed by taking twice the difference between the Log-Likelihood of the full model with more parameters versus the Log-Likelihood of the more parsimonious model. Phenotypic twin correlations, means and variances were extracted from the saturated model. Based on the pattern of phenotypic correlations between MZ and DZ twin pairs, we fitted either an ACE or an ADE model from two groups (MZ and DZ same-sex), with sex and age as a covariate. Relative sub-models (i.e., AE, CE, E) were tested against the full ACE model after the latter was tested against the saturated model. The significance of the CTD informed models was evaluated similarly to that described above. We repeated the procedure as mentioned above for taste-typicality and evaluation-bias for each domain. Narrow-sense heritability ($h^2$) was calculated as the proportion of the variance explained by the A component

relative to the total phenotypic variance:

$$h^2 = \frac{\sigma_A^2}{\sigma_A^2 + \sigma_{C|D}^2 + \sigma_E^2}$$

Likelihood-based 95% Confidence Intervals (CI) around the estimates were calculated using the mxCI OpenMx function (see https://www.rdocumentation.org/packages/OpenMx/versions/2.18.1/topics/mxCI for details; see Supplementary Data 3 for a comprehensive summary).

*Multivariate modelling.* Multivariate modelling exploits patterns of cross-trait cross-twin covariances to partition phenotypic covariance between any two traits into bivariate A, C and E components[80]. Within the multivariate model, it is also possible to compute genetic correlations ($\rho A$), which test for overlapping genetic influences (e.g., pleiotropy) on variation in two traits. We first performed fitting for four saturated plus four full multivariate models, one per aesthetic measure per sample. Two saturated plus two multivariate models were fitted to the taste-typicality and the evaluation-bias data from the discovery sample after regressing out sex and age. Similarly, two saturated plus two multivariate models were fitted to the taste-typicality and the evaluation-bias data from the validation sample after regressing out sex and age. Multivariate models were adapted from the script of Hermine Maes, following Meike Bartel's Boulder International Behaviour Genetic Workshop tutorial, available at https://ibg.colorado.edu/cdrom2020/maes/multivariateScripts/mulACEvc.R. To reduce parameter bias estimation, variance-covariance components were estimated directly by the direct symmetric approach[58]. Fit statistics were evaluated as described for the univariate models. We additionally tested whether the specified models provided a good fit to the data by comparing first the full multivariate ACE (21 parameters = 3 means, 9 variance components, 9 covariance components) with the full multivariate saturated model and then testing the nested multivariate sub-models against the full ACE. Genetic correlations were computed as:

$$\rho A = \frac{\sigma_{A1A2}}{\sqrt{\sigma_{A1}^2 \sigma_{A2}^2}}$$

With $\sigma_{A1A2}$ being equal to the genetic covariance, and $\sigma_{A1}^2$ and $\sigma_{A2}^2$ representing the amount of variance in the traits 1 and 2 (e.g., scenes and faces) explained by additive genetic factors, respectively. CIs around the estimates were calculated as reported above.

*Sensitivity analysis.* We obtained typicality and bias scores in the validation sample as the *mm2* and the within-participant mean of a non-aesthetic task. In the non-aesthetic task, originally administered by Sutherland et al.[45] in the same individuals on the same images, twins rated the perceived dominance of faces. We named the typicality and the bias scores obtained from such dominance ratings as the control-typicality and control-bias scores. We assume any relationship between taste-typicality and control-typicality and evaluation-bias and control-bias to be a confounder (i.e., the overall tendency to display typical judgments and the participant's tendency to give systematically higher or lower ratings regardless of the task). Accordingly, we regressed the taste-typicality and evaluation-bias scores, additionally to individuals' sex and age, on the Fisher z transformed control-typicality and the raw control-bias scores, and re-ran twin and phenotypic correlation analysis, and univariate and multivariate CTD on the residuals of such regressions. We avoided regressing potentially meaningful aesthetic relationships by regressing out control scores obtained from dominance ratings only. Regression was carried out in the statistical package umx[81].

*Ethical approval.* The study by Germine et al.[32] was originally reviewed and approved by the Committee for the Use of Human Subjects at Harvard University and the Australian Twin Registry. The study by Sutherland et al.[45] was originally reviewed and approved by the Human Ethics Committee at the University of Western Australia and at Twins Research Australia. Informed consent was originally obtained by Germine et al. and Sutherland et al. for both studies (see refs. [32,45] for details).

*Simuli availability.* The face impression test has been made available by Sutherland et al.[45] at https://www.testable.org/experiment/855/674205/start. The original pool from which scenes and abstract images were drawn can be found in the supplementary materials of Vessel & Rubin[31] following the link https://jov.arvojournals.org/article.aspx?articleid=2121096#88038837.

**Reporting summary.** Further information on research design is available in the Nature Portfolio Reporting Summary linked to this article.

## Data availability

All data were made available by the original authors of the first two studies, Germine et al.[32] and Sutherland et al.[45], and can be found at https://osf.io/c3hz6/ and https://osf.io/35zf8/?view_only=e76c6755dcea4be2adc5b075cae896e8, respectively. Source data for Figs. 1b and 2 can be found in Supplementary Data 4 and Supplementary Data 5–8, respectively. Source data for Fig. 3, panels a–d, can be found in Supplementary Table 1. Source data for Fig. 3e were obtained from Willoughby et al.[50] Table 2: "Selection of phenotypes from the largest twin meta-analysis to date and mean meta-analytic heritability and shared environment components for each".

## Code availability

We conducted statistical analyses using R in R-Studio, R version 4.2.2, aarch64 architecture. We used the OpenMx https://openmx.ssri.psu.edu/ statistical package version 2.21.1[82], with standard NPSOL and/or CSOLNP optimisers. R markdown files with sections of the results can be found at https://github.com/giacomobignardi/h2_visual_aesthetic_value/tree/main/03_outputs/report. All code used to produce results and unedited figures is available and can be found at https://github.com/giacomobignardi/h2_visual_aesthetic_value/tree/main and is deposited at https://zenodo.org/records/10251279[83].

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

## Acknowledgements

We thank the Australian Twin Registry, Laura Germine et al., and Sutherland et al., for their original work and for making their data openly available. We further thank Laura Germine for their initial input on a previous version of the work and support during revision. We also thank Jeremy B Wilmer for suggestions on estimating sample overlap between the two studies. We also would like to thank the attendees of the 2023 *Comparative Aesthetics: Future Prospects* workshop, held in Vienna, organised by Leonida Fusani, Helmut Leder, Christina Krumpholz and Cliodhna Quigley, for insightful discussions, which improved the revised version of the current manuscript. GB was supported by the German Federal Ministry of Education and Research (BMBF); GB and SEF were supported by the Max Planck Society.

## Author contributions

G.B. and T.J.C.P. conceived the study; G.B. analysed the data; T.J.C.P., E.A.V. and D.J.A.S. validated the work; T.J.C.P. and S.E.F. conceptually validated the work; G.B. visualised the data; G.B. and M.D.T. drafted the manuscript; S.E.F., M.D.T., T.J.C.P., E.A.V., L.F.T. and D.J.A.S. revised the manuscript; T.J.C.P. and E.A.V. supervised the research; all authors reviewed the last version of this manuscript.

## Funding

## Competing interests

The authors declare no competing interests.
