## [Peer Review File · Communications Biology]

Genetic effects on variability in visual aesthetic evaluations are partially shared across visual domainsReviewers' comments:

Reviewer #1 (Remarks to the Author):

Summary

Bignardi and colleagues re-analysed publicly available data to study the genetic and environmental components of taste-typicality and evaluation bias across three subject domains: scenes, faces and abstract images. The authors rightly point to several studies performed on agreement of aesthetic values and do a great job summarising previous results. The methods used are appropriate and I commend the authors for the amount of work and the high-quality visual presentation of their results (figures). Nonetheless, I have some concerns and issues with the manuscript which can be summarised as (i) the lack of (access to?) supplementary figures and tables referenced in the text, (ii) the way in which some results are presented, (iii) the motivation for studying these specific measures of aesthetic values and the lack of further discussion on why common environment seems to play little role with this measured for aesthetic values. Finally and given the broad scope and readership of communications biology, the authors should make a case for the significance of the study and the relevance of their findings early in the article. This currently remains unclear particularly when considering studies on aesthetic values on these samples have been performed and published already. I outline my comments in detail below.

Major:

1. My largest concern with this study concerns to novelty and utility of the results identified. The authors present a great amount of work and results but fall short when discussing the interpretation, implications, and significance of their findings. I would appreciate if the discussion could go deeper into the potential explanations for the results the authors found. Why is taste-typicality different from aesthetic agreement? Which one would be deemed a better measure of "Aesthetic Value"? Why focus on the former rather than the latter? Keep in mind that communications biology will have a broad readership of non-psych, non-cognitive scientists who will not immediately see the value of these findings and may be unfamiliar with the methods and terms presented.
2. It is surprising (and slightly counter intuitive) that the authors find no evidence of common environment for many of their analyses. This makes the results interesting, but discussion on why this may happen, and perhaps a word of caution on what this means (and what this does not mean) is warranted. There's been numerous examples of genetic studies of social and psychological traits that are taken out of context and used to defend racist or xenophobic arguments. Explicitly stating what results do not mean can help prevent that.
3. The authors report AE models to fit better in most cases. I would argue that it is important to report the full ACE models as primary results, with the significance and confidence intervals of each component presented. The authors intended to provide more information regarding the model comparison (and previously the testing of assumptions such as homogeneity of means and variances) in the supplementary materials but I could not find such sections there (I could not find a second supplementary file either). I expect some degree of contribution from the three components and given the sample sizes I would be weary of disregarding the fit and estimates of the ACE model. To be clear, I think the manuscript would benefit of showing full ACE models for both cohorts in a main text table, and then another table with the results for the best model.
4. If the second sample partially overlaps the first one it is important to i) highlight the degree of overlap in the main text (perhaps a venn diagram), and ii) remind the reader in the discussion that the term replication/validation sample is used loosely in this study for practical reasons.
5. I could not find supplementary figure S8, nor table S3, S4 or S5. This is probably an error when uploading the files but it is hard to evaluate the manuscript without the rest of the supplementary material.
6. Without knowing the context of most images (i.e., were visually unappealing scenes showed?) It is hard to make sense of the findings from evaluation bias. The authors describe "Evaluation bias was measured as the individual average of aesthetic ratings per visual domain, reflecting the magnitude of the overall pleasantness experienced by individuals when evaluating visual images". Would such definition not be prone to be confounded by personality and psychological traits not directly related to

aesthetic values but to general agreeableness?

Minor:

1. In the following section:

"The twins had originally rated 65 abstract images in the discovery sample, including a subset of 15 repeated images. They had also rated 65 and 74 images of scenes (including 15 and 24 repeats), and 260 and 150 images of faces (60 and 50 repeats) in the discovery and validation samples, respectively (see Figure 1a and Methods for details)."

Can the authors clarify why there are two sets of values for faces and scenes, but only one of abstract images?

2. I would suggest that figure S1 should include boxplots of intra-rater reliability per section for each of the repeated measures. Exclusion of images with extremely low average reliability should be considered as a sensitivity analysis.

3. Are the results in Table 2 from the main or the replication sample? The text seems to imply it is the replication sample, but a table with the results from the discovery sample are not shown.

4. Do the authors have any hypotheses for the differences observed for abstract vs face taste-typicality. I can think of phenomena such as evolution, the formation of tribes and assortative mating as potential explanations, but I am curious for the authors hypotheses.

Reviewer #2 (Remarks to the Author):

The current study investigates factors that influence aesthetic values across three visual domains: abstract images, scenes, and faces. By analyzing data from monozygotic and dizygotic twins, the authors explore the role of genetic and environmental influences on individual variability in preferences. The results reveal that genetic factors contribute to 25-41% of the variance in taste-typicality and evaluation-bias, with some shared genetic effects across visual domains. However, taste-typicality for abstract images was found to be influenced only by shared and unique environmental factors. Overall, the study demonstrates that both genetic and environmental factors play a role in shaping aesthetic value across distinct visual domains, and that the heritability of major dimensions of aesthetic evaluations is comparable to other complex human traits.

I enjoyed the opportunity to read this paper, wherein the author presents a well written account of results from a methodologically rigorous study. By using multivariate twin models to investigate these phenotypes, the authors can provide a valuable contribution to the literature on the heritability of complex human traits.

Here are some concerns and suggestions for improvement:

1. Figure 2 could be improved for clarity. The twin correlations are difficult to discern, as the yellow-on-white contrast is needlessly taxing. (The viridian color palette may not be optimal for nominal categories). However, I appreciate that the correlations can be found also in the supplementary materials, though why are the lower thresholds given as NA for all that I assume are non-significant?
2. Additionally, panel G is overly detailed, with the inclusion of too many specific medical/somatic phenotypes (metabolic and connective tissues), and not more relevant ones like personality. I find this somewhat distracting. At very least, the three bars with data from the current study could be highlighted (made darker) to stand more out.
3. In table 2 it is unclear why a column is labeled C|D when none of the models include dominance.
4. The rating scale for scenes and faces differs (1-7 vs 1-9), why?
5. In the expression for the genetic correlation on page 20, I believe the "+" in the denominator should be replaced with "*" (also for the legend of figure 4, and same holds for E-correlation).
6. Regarding the mixed models, I struggle to see how you have enough information to estimate the

interaction term. As far as I understand, you need more observations than the number of unique combinations of individual and image for the interaction term to be identified. Is it possible in this case due to the (few) repeated exposures?

7. The reasons for not including taste-typicality correlations for faces could be explained.

8. Although the authors remove data from participants with low test-retest reliability, I cannot find an overall estimate of the reliability of taste-typicality and aesthetic bias. How reliable are these measures? This could potentially have important implications for the estimated heritability, and the claim that "the environment is indeed the leading cause of differences in visual aesthetic evaluations".

9. On the utterly trivial side, could the white "NA" labels in the correlation plots in figure 4 be removed? Why are the two path diagrams in panel a labeled differently? Not convinced the path diagram symbols really add to the clarity.

Overall, I find the paper intriguing and impressively rigorous.

Addressing the minor points above will further strengthen the paper and make it more accessible to readers.

Dear Reviewers,

We are grateful for your comments and the time taken to thoroughly review our manuscript. We have revised our paper based on these comments and answered each below. Please find the original review *comments in italics*, followed by our response. We found the comments and advice we received were of high quality. As such, we took particular care in integrating the comments, which resulted in a majorly revised and substantially improved manuscript. We note that major changes in the manuscript are highlighted in red.

REVIEWER 1
MAJOR

R1.comment1: *My largest concern with this study concerns to novelty and utility of the results identified. The authors present a great amount of work and results but fall short when discussing the interpretation, implications, and significance of their findings. I would appreciate if the discussion could go deeper into the potential explanations for the results the authors found. Why is taste-typicality different from aesthetic agreement? Which one would be deemed a better measure of "Aesthetic Value"? Why focus on the former rather than the latter? Keep in mind that communications biology will have a broad readership of non-psych, non-cognitive scientists who will not immediately see the value of these findings and may be unfamiliar with the methods and terms presented.*

R1.answer1: We thank the reviewer for this important comment. We have now complemented our discussion by expanding the previous interpretation of our results, particularly highlighting the relevance of our findings for the readership of *Communication Biology* and making it more accessible to the general reader. The reviewer can find a new paragraph on heritability, response selection, and the potential interest for evolutionary and comparative studies. Moreover, we believe these above-mentioned and further specific amendments briefly address each of the questions highlighted by the reviewer. Given that we have greatly expanded our interpretation of the results, we point the reviewers to our majorly revised discussion, where we directly address the major concern of the reviewer. [Please see pages 14-17].

R1.comment2: *It is surprising (and slightly counter intuitive) that the authors find no evidence of common environment for many of their analyses. This makes the results interesting, but discussion on why this may happen, and perhaps a word of caution on what this means (and what this does not mean) is warranted. There's been numerous examples of genetic studies of social and psychological traits that are taken out of context and used to defend racist or xenophobic arguments. Explicitly stating what results do not mean can help prevent that.*

R1.answer2: We thank the reviewer for this comment and understand the reviewer's concerns. After expanding our main findings with a more nuanced representation of our results, we discuss in detail why this might be the case.

First, we expand our interpretation to the full ACE models [Please see Table 2]. We show that non-significant C estimates extracted from the full ACE are all between .00 and .07. These results fit well with our original report and with the behavioural genetics landscape of complex traits, which shows that for many behavioural traits, environmental effects account for little inter-individual variance (Polderman et al., 2015, Willoughby et al., 2023). To further contextualize the results within previous behavioural genetics literature, we implement the

changes in an updated figure 2, including C estimates. Moreover, to guide future research on the investigation of shared effects, we also provide simulated power curves for a range of hypothetical C, hoping to raise awareness that only by using massive samples (that we believe to be unlikely to be obtained for studies of this type) can the presence of small C effects (e.g., .07) be properly rejected. Nevertheless, we stress that if present, C effects are smaller than the additive-genetic and non-shared environmental effects, aligning well with the second “law” of behavioural genetics”. [Please see pages 16 and Supplementary 11]

Second, motivated by the reviewer's concern, we propose an additional hypothesis on the possible small/trivial C effects detected in this study. We note that for other cognitive traits, such as general intelligence, shared components within family members contribute to variation in childhood yet tend to progressively decrease until becoming trivial in adulthood (replaced by higher heritability, e.g., the Wilson effect). We suggest that the formation of aesthetic value might follow a similar trend. We discuss this suggestion and leave it open as a question for further studies. [Please see pages 16]

Third, to raise awareness in the naïve reader, we have now included upfront in the introduction a disclaimer on what heritability is and what is not. We explicitly refer the reader to excellent work tackling this timeless and relevant issue (Visscher, Hill, & Wray, 2008). [Please see page 3]

R1.comment3: *The authors report AE models to fit better in most cases. I would argue that it is important to report the full ACE models as primary results, with the significance and confidence intervals of each component presented. The authors intended to provide more information regarding the model comparison (and previously the testing of assumptions such as homogeneity of means and variances) in the supplementary materials but I could not find such sections there (I could not find a second supplementary file either). I expect some degree of contribution from the three components and given the sample sizes I would be weary of disregarding the fit and estimates of the ACE model. To be clear, I think the manuscript would benefit of showing full ACE models for both cohorts in a main text table, and then another table with the results for the best model.*

R1.answer3. We thank the reviewer for this on-point comment. As discussed in answer 2, we now report the ACE models' estimates and power analyses in the main and supplemental texts, respectively. [Please see Table 2 and S11].

We also amended the incorrect submission by correctly submitting the missing Supplementary material. [Please see Supplementary Files SF1 to SF3].

The reviewers can also find all results in the following GitHub repository:

https://github.com/giacomobignardi/h2_visual_aesthetic_value/tree/main/03_outputs/processedData.

R1.comment4: *If the second sample partially overlaps the first one it is important to i) highlight the degree of overlap in the main text (perhaps a venn diagram), and ii) remind the reader in the discussion that the term replication/validation sample is used loosely in this study for practical reasons.*

R1.answer4: We thank the reviewer for this observation, yet regret to inform the reviewer that, due to data sharing regulations, available data were anonymised. To the best of our knowledge, and as confirmed by a brief exchange with some of the authors of the original studies, it is, therefore, not possible to know the overlap between the two samples.

Nevertheless, since the samples are from the same registry, and a conservative number for the number of individuals included in the registry exists (e.g., 40,000 individuals; Hopper et al., 2013), we were able to estimate what the average overlap between any two randomly drawn samples of the size of the Germine et al. and the Sutherland et al. study. This estimate, which calculates an approximate overlap of less than 50 individuals, shows that possible overlaps are of minor concern.

At the same time, we believe this doesn't represent a large issue as the data collection was independent, used a slightly different rating scale, and included different conditions. We stress that the aim of including the replication sample was to show the robustness of the results to possible confounders and ensure that estimates were relatively stable. We have acknowledged this and made it less ambiguous throughout the entire article, and included an additional section in the limitation of this study [Please see page 18].

Furthermore, to partially complement our replication analysis, we also now include a fully independent sample of N = 78 unrelated individuals and validate phenotypic findings, which we believe further strengthen our conclusion and partially address some of the concerns raised by the reviewer. [Please see Supplementary S8 and S9]

R1.comment5: *I could not find supplementary figure S8, nor table S3, S4 or S5. This is probably an error when uploading the files but it is hard to evaluate the manuscript without the rest of the supplementary material.*

R1.answer5: We apologise for the inconvenience. There were some typos in the submitted version. We have now resubmitted the requested material. [Please see Supplementary Files SF1 to SF1, and Supplementary Tables 1 and 2 in the main Supplement] Furthermore, following open access practices pushed all code and outputs to the following GitHub repository https://github.com/giacomobignardi/h2_visual_aesthetic_value/tree/main.

R1.comment6(part1): *Without knowing the context of most images (i.e., were visually unappealing scenes showed?) It is hard to make sense of the findings from evaluation bias.*

R1.answer6(part1): We agree with the reviewer that having more context might increase the interpretability of the study and therefore included the link to the original study in which the scenes and images were originally introduced, with a direct link to the images <https://jov.arvojournals.org/article.aspx?articleid=2121096#88038837>. Furthermore, we also included the link to the original Sutherland et al. online study, allowing readers to experiment with the task <https://www.testable.org/experiment/855/674205/start>. [Please see pages 23-24].

R1.comment6(part2): *The authors describe "Evaluation bias was measured as the individual average of aesthetic ratings per visual domain, reflecting the magnitude of the overall pleasantness experienced by individuals when evaluating visual images". Would such definition not be prone to be confounded by personality and psychological traits not directly related to aesthetic values but to general agreeableness?*

R1.comment6(part2): Regarding the confounding effect, our analysis and sensitivity analysis show that general psychological traits do not likely confound evaluation. We explain below why we believe this to be the case.

We regressed general bias and general typicality from the evaluation bias and taste-typicality scores in the sensitivity analysis. This analysis used a non-aesthetic judgment (judging the dominance of faces). This step discounted some of the personality and general psychological effects over the general tendency to rate high/low and to display ratings similar to the group. [Please page 14-15]

Nevertheless, we acknowledge that it is very difficult (perhaps impossible) to discount all possible confounders and that other psychological traits might relate to the aesthetic-value-specific heritability estimates. We acknowledge this in the discussion and point to it as an interesting avenue for future research [Please see page 15].

MINOR

R1.comment7: *In the following section:*

“The twins had originally rated 65 abstract images in the discovery sample, including a subset of 15 repeated images. They had also rated 65 and 74 images of scenes (including 15 and 24 repeats), and 260 and 150 images of faces (60 and 50 repeats) in the discovery and validation samples, respectively (see Figure 1a and Methods for details).” Can the authors clarify why there are two sets of values for faces and scenes, but only one of abstract images?

R1.answer7: This is entirely due to the decision made by the original research group. While Germine et al. collected abstract images as an additional supplement to their work on the attractiveness of faces, Sutherland et al. did not include abstract images in their work on face impressions.

R1.comment8: *I would suggest that figure S1 should include boxplots of intra-rater reliability per section for each of the repeated measures. Exclusion of images with extremely low average reliability should be considered as a sensitivity analysis.*

R1.answer8: We thank the reviewer for this comment and append the requested box plot in the supplement. [Please see Supplementary S1 and Figure S2]. We note that, as there are no images with inter-image reliability of less than .5, extra sensitivity analyses are not needed.

R1.comment9: *Are the results in Table 2 from the main or the replication sample? The text seems to imply it is the replication sample, but a table with the results from the discovery sample are not shown.*

R1.answer9: Results in Table 2 are reported for both samples. We have now made it clearer that this is indeed the case [Please see Table 2]

R1.comment10. *Do the authors have any hypotheses for the differences observed for abstract vs face taste-typicality. I can think of phenomena such as evolution, the formation of tribes and assortative mating as potential explanations, but I am curious for the authors hypotheses.*

R1.answer10: We are glad that the reviewer themselves showed some curiosity about this novel result. We now discuss the relevance of this difference for evolutionary and comparative studies of aesthetics [Please see page 15]. Furthermore, we have contextualized these results within a recently proposed framework where genetic and environmental effects differently constrain the aesthetic evaluation of natural and non-natural kinds. [Please see pages 15-16].

REVIEWER 2

R2.comment1: *Figure 2 could be improved for clarity. The twin correlations are difficult to discern, as the yellow-on-white contrast is needlessly taxing. (The viridian color palette may not be optimal for nominal categories). However, I appreciate that the correlations can be found also in the supplementary materials, though why are the lower thresholds given as NA for all that I assume are non-significant?*

R2.answer1: We thank the reviewer for this suggestion. We have now adapted Figure 2 to improve clarity. Furthermore, we provided unbounded CI estimates for all models. To be clear, an NA was present as a lower boundary only when the CI was smaller than 0 and, as correctly pointed out by the reviewer, indicated that the correlation was not significant. [Please see Figure 2]

R2.comment2: *Additionally, panel G is overly detailed, with the inclusion of too many specific medical/somatic phenotypes (metabolic and connective tissues), and not more relevant ones like personality. I find this somewhat distracting. At very least, the three bars with data from the current study could be highlighted (made darker) to stand more out.*

R2.answer2: We are grateful for this suggestion. We have now stratified our results within the updated primer from Willoughby et al., 2023, focusing on *Psychiatric, Cognitive and related traits and personality, beliefs and attitudes* categories. [Please see Figure 2]

R2.comment3: *In table 2 it is unclear why a column is labeled C|D when none of the models include dominance.*

R2.answer3: Full models were informed by comparing MZ and DZ twin correlations. Some full models included expected dominance, and some did not. Although we originally reported the final AE model, we now implemented a table with the full ACE or ADE model. We believe this implementation will resolve the reviewer's concern, as it should improve clarity. [Please see Table 2]

R2.comment4: *The rating scale for scenes and faces differs (1-7 vs 1-9), why?*

R2.answer4: Germine et al., used a 1-7 scale, while Sutherland et al. used a 1-9 scale. Unfortunately, we do not know why the latter authors opted for a 1-9 instead of a 1-7, as this information was not provided in the original work.

R2.comment5: *In the expression for the genetic correlation on page 20, I believe the "+" in the denominator should be replaced with "*" (also for the legend of Figure 4, and the same holds for E-correlation).*

R2.answer5: We are grateful, and thank you for spotting this typo. Although we correctly computed both additive-genetic and environmental correlations, we indeed introduced a typo in the formula reported in the manuscript. We now have amended the formula correctly. [Please see page 23]

R2.comment6: *Regarding the mixed models, I struggle to see how you have enough information to estimate the interaction term. As far as I understand, you need more observations than the number of unique combinations of individual and image for the interaction term to be identified. Is it possible in this case due to the (few) repeated exposures?*

R2.answer6: We understand the reviewer's concern, as this might not be apparent. The variance component analysis, initially proposed by Hehman et al., (2017; JPSP), later applied by Sutherland et al. (2020; PNAS), applies a (null) intercept-only model with individuals and images as random effects. As such, two repeated exposures are sufficient to allow the model to obtain an overall estimate for the interaction term, which can be simply thought of as the amount of repeatable variance not due to within-individual average ratings and shared variance across images. We point further to Martinez, Funk, and Todorov (2020; Behav Res Methods), where extensive simulation studies highlight the strengths (and the limitations) of this method within our current implementation.

R2.comment7: *The reasons for not including taste-typicality correlations for faces could be explained.*

R2.answer7: We note that the correlations between taste-typicality for faces are already included in Figure 3, X-axis.

R2.comment8: *Although the authors remove data from participants with low test-retest reliability, I cannot find an overall estimate of the reliability of taste-typicality and aesthetic bias. How reliable are these measures? This could potentially have important implications for the estimated heritability, and the claim that "the environment is indeed the leading cause of differences in visual aesthetic evaluations".*

R2.answer8: We thank the reviewer for this very welcomed suggestion. We have now included test-retest reliability for taste-typicality and evaluation bias based on the repeated images. Moreover, we have also included an additional sample of N = 78 individuals based on Sutherland et al. data and computed three-day test-retest reliability for evaluation bias and taste typicality. We show that the test-retest of this metric is good to excellent both within and between days. We included details on the sample from which and the methods to compute intra-rater reliability. [Please see Supplement S7 and S8, the amendment in the main text can be found on page 8]. These results indicate that the E component is minimally confounded by error variance.

R2.comment9: *On the utterly trivial side, could the white "NA" labels in the correlation plots in figure 4 be removed? Why are the two path diagrams in panel a labeled differently? Not convinced the path diagram symbols really add to the clarity.*

R2.answer9: We thank the reviewer for this last comment and amended Figure 2 to increase clarity.

REVIEWERS' COMMENTS:

Reviewer #1 (Remarks to the Author):

The authors have addressed all of my comments and concerns.

Reviewer #2 (Remarks to the Author):

I have now been able to review the revised manuscript, and I am pleased to say that all of my very small initial concerns have been addressed.

The modifications made to Figure 2, I believe, have enhanced its clarity, especially by comparing taste-typicality and evaluation-bias with more relevant traits in panel G.

In figure 4, I think including the estimates rather than the labels in the path diagrams makes the figure more readable. I did however think the shaded boxes in the previous version were helpful, so my comment was really only on the "NA" labels. However, this is a trivial point, and the figure looks very good.

The detailed explanation provided regarding the estimation of the interaction term in the mixed models has also alleviated my concerns.

Again, this is a very interesting and methodologically rigorous paper, so thank you for the opportunity to review it.